# EMBEDDING INTERPRETABILITY SCORE: A DOMAIN-AGNOSTIC REPRESENTATION QUALITY ASSESSMENT

## ABSTRACT

Interpretability has long influenced the selection of machine learning models, yet the role of data representations remains relatively underexplored. Although model choice is known to influence performance, the interpretability of embedding models can also be equally critical. In this study, we present a comparative analysis of various black-box and interpretable embedding models within multiple domains, including natural language processing and computer vision. We introduce a domain-agnostic quantitative score called *Embedding Interpretability Score (EIS)* to measure the interpretability of embedding models based on three fundamental properties: dimensionality, which reflects representational compactness; sparsity, which highlights feature selectivity; and clusterability, which measures semantic organization. Our results indicate that, in general, the choice of the embedding technique exerts a significant influence on downstream performance in comparison to classifier selection. Interestingly, the relationship between interpretability and performance differs across modalities: in NLP tasks, higher-performing embeddings tend to have lower interpretability, whereas in CV tasks, embeddings with higher interpretability often achieve better downstream performance.

## 1 INTRODUCTION

Machine learning models have become integral to society, shaping critical aspects of daily life such as employment, credit, education, and legal decision-making (Dixon et al., 2020; Nietzel, 2022; Barocas & Selbst, 2016; Habehh & Gohel, 2021). While black-box models have gained significant popularity for their ability to uncover complex patterns in data, their lack of transparency raises concerns regarding trust, accountability, and responsible decision-making (Rudin, 2019; Lipton, 2018; Lakkaraju et al., 2016; Mishra et al., 2021). Consequently, there has been considerable debate regarding the use of black-box versus *inherently* interpretable models (Rudin, 2019; Liao et al., 2024; Ghasemi et al., 2023; Atrey et al., 2025). Yet, comparatively little attention has been paid to the interpretability of data representations in machine learning.

Modern machine learning systems often rely on state-of-the-art embedding models to encode data into informative representations (Devlin et al., 2019; Caron et al., 2021; Liu et al., 2019; Mikolov et al., 2013). These embeddings transform raw inputs into vector representations that capture semantic or structural information, enabling downstream tasks such as classification, retrieval, and generation. However, while the choice of model is known to impact performance, the interpretability of these embeddings themselves remains underexplored. Less complex embedding models, such as Word2Vec (Mikolov et al., 2013) or GloVe (Pennington et al., 2014), are often more interpretable, as their representations are simpler and easier for humans to relate to semantic meaning. In contrast, more complex embeddings, such as those produced by BERT (Devlin et al., 2019), typically achieve higher task performance but are less transparent, making it challenging to understand how they encode and organize information. Understanding which embeddings are more interpretable can provide insights into the transparency and trustworthiness of the overall system, guiding both model selection and deployment.

*Main Contributions:* Towards addressing this challenge, in this work, we propose a domain-agnostic methodology to measure the interpretability of embedding models. Our proposed interpretability score — which we call the **Embedding Interpretability Score (EIS)** — takes into account three fundamental properties: dimensionality, sparsity, and clusterability. We demonstrate the use of the

interpretability score on a diverse range of embedding models across classification tasks within multiple domains, including natural language processing (NLP) and computer vision (CV). Additionally, we analyze the trade-off between embedding interpretability and performance, showing that while less interpretable embeddings often achieve higher accuracy, interpretable embeddings can remain competitive and provide greater transparency. Our results highlight that embedding model choice can even have a greater impact on downstream performance than classifier selection, underscoring the importance of considering interpretability in representation learning. Importantly, we reveal a striking domain-specific trend: in NLP tasks, embeddings with higher predictive accuracy tend to be less interpretable, whereas in CV tasks, embeddings that are more interpretable often achieve superior performance. To summarize, our main contributions are:

- We propose a domain-agnostic score to quantify embedding interpretability – that we call *Embedding Interpretability Score (EIS)* – relying on dimensionality, sparsity, and clusterability.

- We evaluate EIS across a diverse set of embedding models in multiple domains, including NLP and CV, demonstrating its practical utility for comparing embeddings.

- We reveal an interesting domain-specific trend: in NLP, embeddings with higher predictive accuracy tend to be less interpretable, whereas in CV, embeddings that are more interpretable often achieve superior performance.

Our results underscore that embedding choice can have a larger impact on downstream performance than classifier selection, emphasizing the importance of interpretability in representation learning. Furthermore, our approach enables pairwise comparisons and ordering among different embedding models based on their relative *degree* of interpretability, rather than just placing them in the traditional glass-box or black-box categories.

## 2 RELATED WORK

Prior work has aimed to design embeddings with dimensions that are directly interpretable. For instance, Trifonov et al. (2018) introduced sparse sentence embeddings and a novel automated metric based on topic coherence to quantify interpretability, showing that sparsity can improve semantic transparency without sacrificing performance. Similarly, Senel et al. (2018) analyzed semantic structure in word embeddings and also leveraged word category datasets to quantitatively assess interpretability, highlighting the potential for dataset-driven evaluation methods. Chandrahas et al. (2020) extended these ideas to knowledge graph embeddings by designing knowledge graph embeddings and quantitatively evaluating their semantic coherence. These works focus on designing embeddings that are interpretable by construction, whereas our approach provides a model-agnostic, quantitative framework to evaluate interpretability across multiple embedding types and domains. These works emphasize designing embeddings that are interpretable by construction.

Beyond construction, several studies have focused on systematically assessing interpretability. Fang et al. (2022) examined the construct validity of text embeddings in survey contexts, while Santis et al. (2025) proposed linearly-interpretable concept embeddings that explicitly link dimensions to semantic categories. Sun et al. (2025) introduced CQG-MBQA, a framework for producing interpretable semantic text embeddings, in which each dimension corresponds to answers to low-cognitive-load binary questions, making the embedding space inherently human-interpretable. Moreover, Opitz et al. (2025) outlined principles for designing and evaluating interpretable embeddings, emphasizing sparsity, semantic alignment, and the explainability of similarity relationships. Together, these works underscore the value of quantitative evaluation.

In addition to embedding interpretability, previous studies have examined the general trade-off between model interpretability and accuracy. Rudin (2019) argues that black-box models often lead to unreliable outcomes in high-stakes decisions, advocating for inherently interpretable models that improve transparency and trust. Atrey et al. (2025) conducted a comparative study of black-box and interpretable models in the task of predicting product ratings from reviews. They introduced a Composite Interpretability (CI) score that incorporates expert assessments of simplicity, transparency, and explainability to systematically rank models by interpretability.

While prior work has focused on designing embeddings with interpretable dimensions or optimizing individual properties such as sparsity or clusterability, our approach stands out in several ways.

It combines multiple interpretability properties into a single score, providing a holistic assessment rather than evaluating each property in isolation. The method is domain-agnostic, implying that is applicable across multiple domains such as NLP and CV. Additionally, the method can be validated empirically through measurable embedding properties and downstream task performance, demonstrating its meaningfulness without relying on expert assessments. Importantly, *our methodology provides a pathway for ranking embedding models by their degree of interpretability, going beyond the traditional dichotomy of glass-box versus black-box models.*

## 3 PROPOSED INTERPRETABILITY SCORE

We propose a domain-agnostic methodology for quantifying the interpretability of embedding models, independent of the downstream classifier. Our approach integrates three fundamental criteria—**Dimensionality**, **Sparsity**, and **Clusterability**—each normalized to a fixed reference scale and aggregated into an embedding interpretability score. By deriving sparsity and clusterability from a pooled set combining all test examples across the datasets within each domain, the methodology ensures reproducible, dataset-aware comparisons without biasing toward a single benchmark. In contrast, dimensionality is an intrinsic property of the embedding architecture and does not depend on the data. Taken together, the proposed score balances these criteria without prioritizing any single one, enabling consistent evaluation of embedding models within diverse domains (e.g., NLP, CV, and multi-modal tasks).

**Dimensionality.** Embedding dimensionality refers to the size of the vector representing each data point in the embedding space. Lower-dimensional embeddings are generally easier to visualize and reason about, whereas higher-dimensional embeddings can encode more complex patterns at the cost of interpretability. To quantify this effect, we compute the log-scaled dimensionality of each embedding and normalize it to a fixed reference range (e.g., 64 to 2048), as shown in Equation 1. This normalization ensures comparability across models with widely varying representational sizes.

$$D_e = \frac{\log_{10}(d_{\max}) - \log_{10}(d_e)}{\log_{10}(d_{\max}) - \log_{10}(d_{\min})}, \quad \text{with } D_e \in [0.05, 1] \tag{1}$$

where: $D_e$ is the normalized dimensionality for embedding $e$; $d_e$ is the dimensionality of embedding $e$; and $d_{\min}$ and $d_{\max}$ represent the fixed reference range dimensionality (e.g., 64-2048). The range of the fixed reference range has been selected to cover the dimensionalities of nearly all popular embedding models used in prior work, ensuring a consistent comparison across models.

**Sparsity.** We define sparsity using the Hoyer measure (Hoyer, 2004), a threshold-free metric that quantifies the concentration of values in the embedding space. Sparse embeddings are often more interpretable because individual dimensions correspond to fewer, more distinct features. The Hoyer sparsity (originally ranging from $-1$ to 1) is normalized to the $[0.05, 1]$ range, ensuring comparability across embeddings of different dimensionalities. A score of 0.05 corresponds to a maximally dense embedding, while a score of 1 indicates maximal sparsity. To reduce the skewness of the raw Hoyer measure and ensure that differences in sparsity are more evenly reflected across the full range, we apply a square-root transformation. This makes the score more balanced and sensitive across both low and high sparsity values while keeping it normalized to $[0.05, 1]$. The normalized sparsity score for an embedding $e$ is defined as:

$$S_e = \left( \frac{\sqrt{d_e} - \frac{\|x\|_1}{\|x\|_2}}{\sqrt{d_e} - 1} \right)^{\frac{1}{2}}, \quad \text{with } S_e \in [0.05, 1] \tag{2}$$

where: $S_e$ is the Hoyer sparsity score for embedding $e$; $d_e$ is the dimensionality of the embedding $e$; $\mathbf{x}$ is the embedding vector; $\|\mathbf{x}\|_1$ is the L1 norm; and $\|\mathbf{x}\|_2$ is the L2 norm.

**Clusterability.** Embedding clusterability reflects how effectively embeddings capture discrete semantic or visual concepts. We approximate this property using the silhouette coefficient, which quantifies both cohesion (within-cluster compactness) and separation (between-cluster distance) (Rousseeuw, 1987). Values closer to 1 indicate well-separated, coherent clusters, suggesting the embedding organizes concepts in a human-aligned manner. Distances are measured using cosine similarity, and we apply k-means clustering with the number of clusters equal to the number of target classes in the dataset (e.g., five clusters for a five-class task). To ensure comparability with other

criteria, we normalize the silhouette score into the $[0, 1]$ range, as shown in Equation 3.

$$SS = \frac{1}{N} \sum_{i=1}^{N} \frac{b(i) - a(i)}{\max(a(i), b(i))}, \quad C_e = \frac{SS + 1}{2} \in [0.05, 1] \tag{3}$$

where: $SS$ is the silhouette score; $N$ is the number of points in the reference dataset; $a(i)$ is the average distance between $i$ and all other points in its cluster; $b(i)$ is the minimum average distance from $i$ to points in other clusters; and $C_e$ is the normalized clusterability score for embedding $e$. To ensure alignment with the downstream task, the number of clusters used in the silhouette score is set equal to the number of target classes in the dataset. For example, in our classification setting with five rating levels, we use five clusters. This choice grounds the clusterability score in the task-specific structure while still reflecting general representational properties of the embeddings.

The final EIS is defined as the simple geometric mean of the three normalized components: dimensionality, sparsity, and clusterability, as shown in Equation 4.

$$\text{EI}_e = (D_e \cdot S_e \cdot C_e)^{\frac{1}{3}}, \tag{4}$$

where $D_e$, $S_e$, and $C_e$ denote the normalized dimensionality, sparsity, and clusterability scores, respectively. The geometric mean ensures that each aspect meaningfully influences the overall score: low values strongly reduce it, while high values contribute proportionally, producing a balanced, transparent, and reproducible measure of embedding interpretability. While we have demonstrated the use of EIS on NLP and CV in this study, the framework is broadly applicable to other modalities such as speech and even multi-modal domains.

## 4 EXPERIMENTAL SETUP

### 4.1 DATASETS

To evaluate the proposed interpretability score across diverse representation learning settings, we employ datasets from both NLP and CV. This enables us to examine whether the interpretability framework yields consistent and meaningful results within each domain, while avoiding direct comparisons across modalities.

Our NLP datasets are retrieved from a larger database consisting of a large crawl of product reviews from Amazon (Ni et al., 2019). The database contains 82.83 million unique reviews from approximately 20 million users. The reviews are in text format while the ratings are in numerical format ranging from 1 to 5. To analyze product reviews and ratings, 15,000 product reviews and ratings have been extracted from the following three product categories leading to three datasets: (i) Cell Phones and Accessories (**CPA**); (ii) Electronics (**ET**); and (iii) Video Games (**VG**). To ensure a balanced dataset, each category consists of 1,000 reviews and ratings grouped by each rating star (1-5), consisting of a total of 5,000 reviews. Considering three product categories enables us to assess the robustness and generalizability of results across varied product contexts.

Our CV datasets are retrieved from three publicly available sources: **CIFAR-10**, **STL-10**, and **Fashion-MNIST**. CIFAR-10 consists of 60,000 color images of size $32 \times 32$ across ten object classes, with 50,000 training and 10,000 test examples (Krizhevsky, 2009). STL-10 contains 5,000 labeled training images and 8,000 test images at a higher resolution of $96 \times 96$, also spanning ten object categories (Coates et al., 2011). Fashion-MNIST is a modern drop-in replacement for MNIST, comprising 70,000 grayscale images of size $28 \times 28$ depicting clothing items from ten categories, split into 60,000 training and 10,000 test examples (Xiao et al., 2017). To maintain consistency in evaluation, we use a stratified subset of 5,000 images per dataset, ensuring an equal number of samples across all classes. Using these datasets allows us to test embedding and classifier performance across both simple and complex visual tasks, demonstrating the generalizability of our approach in computer vision contexts.

### 4.2 EMBEDDING MODELS

We evaluate a diverse set of NLP and CV embedding models to study their interpretability and downstream classification performance. The NLP embeddings include Word2Vec (**W2Vec**), which

produces static word embeddings using co-occurrence statistics (Mikolov et al., 2013); **MiniLM**, a compact transformer model designed for efficient contextualized representations (Wang et al., 2020); **MiniLM-100d**, a lower-dimensional variant of MiniLM reduced to 100 dimensions using principal component analysis; **Para-MiniLM**, a MiniLM model fine-tuned for semantic textual similarity tasks; **MPNet**, a transformer that incorporates masked and permuted language modeling (Song et al., 2020); **BERT**, a large transformer pre-trained on masked language modeling (Devlin et al., 2019); **DistilRoBERTa**, a distilled version of RoBERTa with reduced size; **RoBERTa**, a large transformer pre-trained on extensive textual corpora (Liu et al., 2019); and **SetFit-DU**, a fine-tuned embedding trained on Amazon product reviews and ratings to improve task-specific representations.

The computer vision (CV) embeddings include **CLIP-ViT-B32**, a vision transformer trained with contrastive language-image pretraining to produce aligned multimodal embeddings (Radford et al., 2021); **DINO-ViT-B16**, a self-distillation approach for learning vision transformer embeddings without labels (Caron et al., 2021); **ViT-B16**, a standard vision transformer pre-trained on ImageNet for image classification (Dosovitskiy et al., 2021); **DenseNet-121**, a convolutional neural network (CNN) with dense connectivity to improve feature reuse (Huang et al., 2017); **EffNet-B0**, an efficient CNN model optimized for parameter efficiency and accuracy (Tan & Le, 2019); and **ResNet-50**, a residual CNN architecture that enables very deep networks through skip connections (He et al., 2016). We evaluate both transformer-based and CNN-based embeddings to explore effects of model architecture and training methodology on interpretability and task performance. Further details about the models are provided in the Appendix.

### 4.3 Classification Models

To evaluate embedding quality in downstream tasks, we consider a diverse set of classification models spanning linear, ensemble tree, and neural approaches. Linear models include Logistic Regression (**LR**) and Linear Support Vector Machines (**SVMs**), which provide strong baseline performance and are highly interpretable (Pedregosa et al., 2011). Ensemble models include Random Forest (**RF**) (Breiman, 2001), XGBoost (**XGB**) (Chen & Guestrin, 2016), and LightGBM (**LGBM**) (Ke et al., 2017), which leverage multiple weak learners to improve predictive performance and capture non-linear interactions. Finally, neural classifiers include a Multi-Layer Perceptron (**MLP**) and a Keras-based Deep Neural Network (**DNN**) (Pedregosa et al., 2011; Chollet, 2015), which can model complex patterns in high-dimensional embedding spaces. The MLP consists of two hidden layers with 256 and 128 neurons, ReLU activation, and is trained using the Adam optimizer. The DNN consists of two fully connected hidden layers with 256 and 128 neurons respectively, each followed by a dropout layer with a rate of 0.3, and a final output layer with softmax activation for classification. It is compiled with the Adam optimizer and sparse categorical cross-entropy loss, and trained for 10 epochs with a batch size of 64. All models are trained and evaluated on a 70/30 stratified train/test split for each dataset. These classification models are applied to both NLP and computer vision embeddings, providing a consistent framework to assess embedding quality within each modality.

## 5 Results

### 5.1 Computing Embedding Interpretability Score Across Domains

**Embedding Interpretability Scores (EIS):** We compare the interpretability of various NLP and CV embedding models using our proposed EIS, as shown in Table 1. For NLP embeddings, W2Vec and MiniLM achieve the highest EIS, reflecting the benefits of lower-dimensional, compact representations that balance sparsity and clusterability. In contrast, DistilRoBERTa and RoBERTa achieve the lowest EIS, reflecting the hindrance caused by their high dimensionality and dense activations, irrespective of the rich contextual information they capture. Compared to MiniLM, MiniLM-100d achieves a lower EIS; while it benefits from a higher dimensionality score, its inability to capture sparse representations is reflected in the sparsity score, resulting in a lower overall interpretability. Additionally, SetFit-DU exhibits a significantly higher clusterability score compared to other embeddings in the study, which can be attributed to its training on Amazon product reviews and ratings, closely aligning with the review-based datasets used in our evaluation. This demonstrates that task-specific fine-tuning can, in some cases, improve interpretability by producing embeddings that better capture semantically meaningful groupings aligned with the evaluation data. Among CV embeddings, DINO-ViT-B16 and ViT-B16 exhibit the highest EIS, while ResNet-50 scores the

| | Embedding | Actual Dim | Dimensionality | Sparsity | Clusterability | EIS |
|---|---|---|---|---|---|---|
| **NLP** | W2Vec | 300 | 0.554 | 0.925 | 0.533 | 0.649 |
| | MiniLM | 384 | 0.483 | 0.921 | 0.563 | 0.630 |
| | Para-MiniLM | 384 | 0.483 | 0.922 | 0.562 | 0.630 |
| | SetFit-DU | 512 | 0.400 | 0.908 | 0.707 | 0.636 |
| | MiniLM-100 | 100 | 0.871 | 0.417 | 0.544 | 0.583 |
| | MPNet | 768 | 0.283 | 0.931 | 0.573 | 0.532 |
| | BERT | 768 | 0.283 | 0.936 | 0.564 | 0.531 |
| | DistilRoBERTa | 768 | 0.283 | 0.946 | 0.560 | 0.531 |
| | RoBERTa | 1024 | 0.200 | 0.953 | 0.555 | 0.473 |
| **CV** | DINO-ViT-B16 | 768 | 0.283 | 0.890 | 0.564 | 0.522 |
| | ViT-B16 | 768 | 0.283 | 0.799 | 0.563 | 0.503 |
| | CLIP-ViT-B32 | 512 | 0.400 | 0.407 | 0.596 | 0.459 |
| | DenseNet-121 | 1024 | 0.200 | 0.774 | 0.562 | 0.443 |
| | EffNet-B0 | 1280 | 0.136 | 0.722 | 0.545 | 0.376 |
| | ResNet-50 | 2048 | 0.050 | 0.876 | 0.564 | 0.291 |

Table 1: Dimensionality, clusterability and sparsity scores, along with EIS, for NLP and CV embeddings. Higher EIS values indicate embeddings that are more interpretable, reflecting a balance of compactness, sparsity, and cluster structure.

lowest, primarily due to its high dimensionality. Transformer-based embeddings (e.g., ViT-B16, CLIP-ViT-B32, and DINO-ViT-B16) consistently outperform CNN-based embeddings (e.g., ResNet and DenseNet) in interpretability, suggesting that self-attention architectures yield more structured and clusterable representations. Overall, the results demonstrate how embedding characteristics can influence interpretability within multiple modalities.

**Visualization of Embedding Properties:** Figure 1 presents the dimensionality, sparsity, clusterability, and overall EIS for the NLP and CV embeddings. For each embedding, the three fundamental properties are shown as bars, while the EIS is overlaid as a line to highlight the combined interpretability. While the ranges of the sparsity and clusterability scores are relatively similar across embeddings, the EIS helps distinguish their overall interpretability by combining these properties using the geometric mean. The geometric mean ensures that no single property dominates the score and that proportional differences are captured, rather than absolute ones. The figure demonstrates how each property contributes to overall interpretability and emphasizes the trade-offs among dimensionality, sparsity, and clusterability across different embedding models.

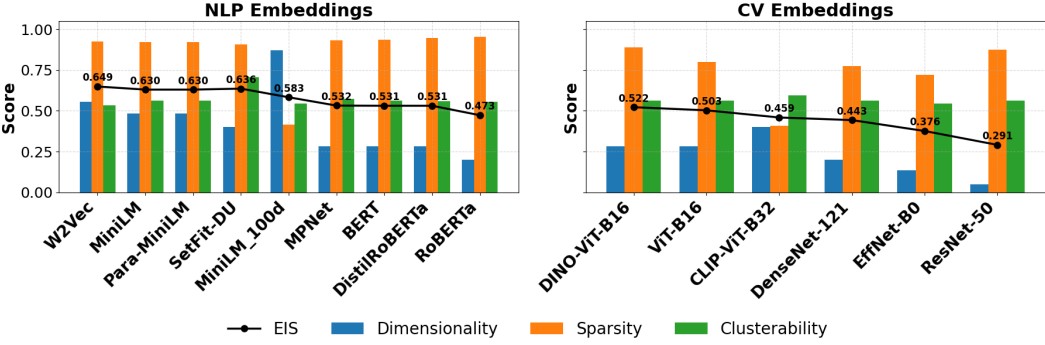

Figure 1: Comparison of embedding interpretability properties (dimensionality, sparsity, and clusterability) across NLP (left) and CV (right) embeddings. The black line with markers shows the overall EIS for each embedding. The figure highlights how each property contributes differently across embeddings, revealing trade-offs between dimensionality, sparsity, and clusterability.

| Dataset | Embedding | Linear Models | | Ensemble Tree Models | | | Neural Classifiers | |
|---|---|---|---|---|---|---|---|---|
| | | **LR** | **SVM** | **RF** | **XGB** | **LGBM** | **MLP** | **DNN** |
| CPA | W2Vec | 38.73% | 39.73% | 36.00% | 38.60% | 39.13% | **40.07%** | 39.53% |
| | MiniLM | 40.13% | 38.53% | 38.53% | 39.33% | 37.67% | 40.00% | **41.40%** |
| | Para-MiniLM | 40.27% | 38.67% | 39.73% | 41.93% | 41.00% | 39.93% | **42.67%** |
| | SetFit-DU | 40.20% | **43.20%** | 38.87% | 39.47% | 39.60% | 41.07% | 38.47% |
| | MiniLM-100 | 37.73% | 38.87% | 36.53% | 35.73% | 35.73% | 39.67% | **39.87%** |
| | MPNet | 46.87% | 46.07% | 40.67% | 43.80% | 45.47% | 45.47% | **47.60%** |
| | BERT | 45.00% | 41.93% | **47.67%** | 45.27% | 45.60% | 43.60% | 45.93% |
| | DistilRoBERTa | 46.07% | 46.87% | 44.27% | 45.80% | 45.53% | 46.13% | **47.13%** |
| | RoBERTa | 50.27% | **51.53%** | 46.53% | 47.13% | 47.53% | 50.07% | 50.73% |
| ET | W2Vec | 44.20% | **46.53%** | 40.93% | 42.53% | 42.53% | 44.33% | 45.13% |
| | MiniLM | 46.40% | 45.53% | 46.13% | 45.40% | 47.00% | 48.47% | **49.00%** |
| | Para-MiniLM | 46.00% | 45.47% | 44.67% | 46.20% | **46.33%** | 44.67% | 45.67% |
| | SetFit-DU | 37.00% | **40.53%** | 37.13% | 39.73% | 39.53% | 39.13% | 37.67% |
| | MiniLM-100 | 46.87% | 46.27% | 43.20% | 44.87% | 45.13% | 46.80% | **47.53%** |
| | MPNet | 51.00% | 50.87% | 47.73% | 50.07% | 50.20% | 50.60% | **53.93%** |
| | BERT | 47.00% | 45.33% | 48.33% | 49.07% | **49.27%** | 47.27% | 48.73% |
| | DistilRoBERTa | 51.53% | 51.40% | 47.73% | 51.93% | 50.80% | 49.93% | **53.33%** |
| | RoBERTa | 54.13% | 53.93% | 51.60% | 52.40% | 53.47% | 54.07% | **54.47%** |
| VG | W2Vec | 41.53% | 41.93% | 39.13% | 41.67% | 41.53% | **43.07%** | 42.07% |
| | MiniLM | 49.33% | 48.40% | 44.47% | 46.20% | 47.60% | **50.80%** | 49.53% |
| | Para-MiniLM | 46.13% | 45.60% | 42.47% | 45.93% | 44.60% | 45.20% | **47.27%** |
| | SetFit-DU | 37.00% | **41.53%** | 39.40% | 40.67% | 40.87% | 39.00% | 37.67% |
| | MiniLM-100 | 48.13% | 49.07% | 42.73% | 46.73% | 45.67% | **50.13%** | 48.80% |
| | MPNet | 52.47% | **53.33%** | 49.07% | 50.20% | 51.47% | 51.53% | 52.73% |
| | BERT | 47.73% | 45.53% | 47.13% | 48.20% | **48.73%** | 47.07% | 47.40% |
| | DistilRoBERTa | 51.60% | 50.27% | 48.60% | 50.27% | 50.87% | 49.40% | 51.53% |
| | RoBERTa | **53.47%** | 51.20% | 50.80% | 53.27% | 53.07% | 50.13% | 51.93% |
| CIFAR-10 | DINO-ViT-B16 | **93.73%** | 92.60% | 89.93% | 90.87% | 91.47% | 93.40% | 93.47% |
| | ViT-B16 | 93.80% | 93.47% | 91.13% | 91.07% | 91.40% | 93.67% | **93.80%** |
| | CLIP-ViT-B32 | 91.93% | 90.67% | 89.67% | 90.47% | 90.87% | 91.13% | **92.33%** |
| | DenseNet-121 | 86.00% | 82.40% | 80.00% | 84.20% | 84.67% | **86.67%** | 86.47% |
| | EffNet-B0 | 87.80% | 86.80% | 84.67% | 85.20% | 85.47% | **88.60%** | 88.00% |
| | ResNet-50 | 85.33% | 84.93% | 84.13% | 84.47% | 85.27% | 85.80% | **86.80%** |
| STL-10 | DINO-ViT-B16 | **99.40%** | 99.13% | 98.60% | 97.53% | 98.13% | 99.07% | 98.93% |
| | ViT-B16 | **99.20%** | 98.93% | 98.27% | 97.60% | 98.40% | 99.07% | 98.93% |
| | CLIP-ViT-B32 | 98.80% | 98.60% | 98.73% | 97.73% | 98.20% | **98.87%** | 98.40% |
| | DenseNet-121 | **97.40%** | 96.80% | 96.07% | 95.13% | 95.87% | 97.13% | 97.20% |
| | EffNet-B0 | 97.80% | **97.93%** | 96.87% | 96.40% | 97.13% | 97.67% | 97.53% |
| | ResNet-50 | 97.53% | 97.27% | 97.40% | 96.40% | 96.33% | 97.53% | **97.73%** |
| Fashion-MNIST | DINO-ViT-B16 | 86.27% | 85.60% | 83.87% | 86.00% | 86.47% | 87.93% | **88.13%** |
| | ViT-B16 | 84.07% | 83.20% | 82.80% | 84.93% | 84.73% | 85.00% | **85.60%** |
| | CLIP-ViT-B32 | **86.93%** | 85.53% | 83.33% | 85.00% | 85.47% | 87.40% | 85.80% |
| | DenseNet-121 | 84.27% | 82.13% | 84.40% | 86.00% | **87.20%** | 85.20% | 85.67% |
| | EffNet-B0 | 87.13% | 85.13% | 84.87% | 85.93% | 86.40% | **87.80%** | 87.00% |
| | ResNet-50 | 84.40% | 84.73% | 83.27% | **85.00%** | 84.60% | **85.00%** | 84.20% |

Table 2: Classification accuracy (%) across embeddings and datasets. While neural classifiers (MLP, DNN) usually achieve the strongest performance, linear classifiers at times outperform them, and ensemble tree models generally underperform. Across both NLP and CV tasks, downstream performance is primarily driven by embedding quality, with less interpretable embeddings often yielding higher accuracy in NLP, whereas more interpretable embeddings tend to perform better in CV.

## 5.2 ACCURACY-INTERPRETABILITY TRADE-OFF

To assess the accuracy–interpretability trade-off, we evaluate a range of embedding models using multiple classifiers on multiple NLP and CV datasets. Table 2 summarizes classification performance across the datasets. Accuracies typically range between 37–54% for the NLP datasets and 82–99% for the CV datasets. Several consistent trends emerge, providing insights into how interpretability relates to predictive accuracy.

**NLP Datasets:** Across all three NLP datasets, traditional embeddings such as Word2Vec consistently underperform transformer-based embeddings. For instance, across the three NLP datasets, RoBERTa achieves accuracies that are on average about 10 percentage points higher than Word2Vec, highlighting the substantial improvement provided by transformer-based representations. Distil-RoBERTa and MPNet also achieve strong performance, typically reaching 51–53% across ET and VG, indicating their robustness across datasets. In contrast, SetFit-DU performs substantially worse,

particularly on ET and VG (37–41%), suggesting that its representations do not generalize well under this evaluation protocol. The spread in accuracy across embeddings is notable, with a gap of nearly 17 percentage points between the weakest and strongest representations. Classifier choice exerts only a modest effect: while MLP and DNN yield small gains (1–2%) over linear baselines, ensemble tree models consistently underperform relative to linear and neural approaches, implying that the embeddings are largely linearly separable. Notably, the embeddings with the highest accuracies, such as RoBERTa and MPNet, also have higher dimensionality and lower sparsity, resulting in lower EIS scores. *This pattern illustrates a clear accuracy–interpretability trade-off, where performance gains come at the expense of interpretability.* Taken together, these results highlight that downstream performance in NLP tasks is driven far more by embedding choice than classifier capacity, with the most accurate embeddings often being the least interpretable due to their higher dimensionality and density.

**CV Datasets:** Across all CV datasets, transformer and CNN-based embeddings exhibit substantial variation in performance. ViT-B16 and DINO-ViT-B16 consistently perform well within each dataset, while embeddings with higher dimensionality and lower sparsity, such as ResNet-50 or EffNet-B0, generally underperform relative to more compact representations. CLIP-B32 performs well overall but shows slightly lower accuracy on MNIST (85–87%), suggesting some sensitivity to dataset characteristics. Similar to NLP, classifier choice has a limited effect: While neural classifiers generally achieve the highest accuracies, there are instances where linear and ensemble tree models outperform them. *Importantly, these observations reveal a striking domain-specific trend: in CV, higher interpretability often aligns with better predictive performance.*

**Visualization of Performance Trends:** To visualize these trends more clearly, Figure 2 presents heatmaps of classifier performance across embeddings for each dataset, ordered by decreasing EIS. The heatmaps reveal that classifier choice has minimal effect: no model consistently outperforms others across embeddings, and differences are generally modest. Meanwhile, the relationship between interpretability and accuracy is domain-specific: in NLP datasets, embeddings with lower EIS tend to achieve higher accuracy, whereas in CV datasets, embeddings with higher EIS often perform better. This figure provides a concise visual summary of these patterns and highlights the central role of embedding quality in downstream performance.

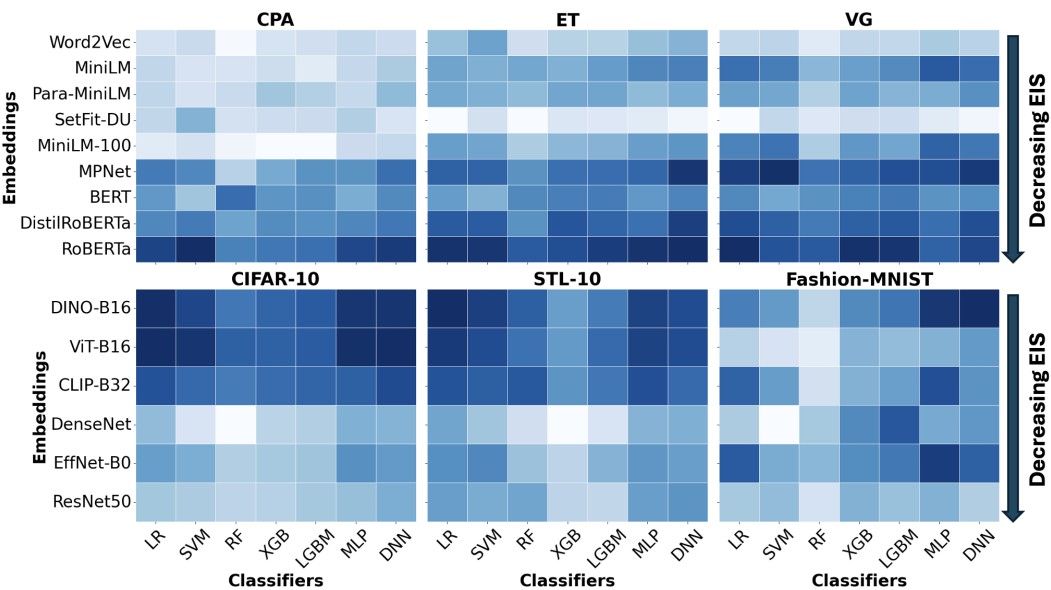

Figure 2: Classification accuracy heatmaps for NLP (top row) and CV (bottom row) embeddings across classifiers, with embeddings ordered by decreasing EIS. For NLP, performance generally improves as interpretability decreases, while in CV, more interpretable embeddings tend to yield higher accuracy. No consistent trend is observed across different classifiers, indicating that embedding choice drives downstream performance more than classifier selection.

**Correlation Analysis:** Table 3 summarizes the Pearson's correlations (Pearson, 1896) between downstream classification accuracy and various embedding interpretability measures, including the overall EIS score, dimensionality, sparsity, and clusterability. Pearson's correlation ranges from -1 to 1, where 1 indicates a perfect positive correlation, 0 denotes no correlation, and -1 represents a perfect negative correlation. For NLP datasets (CPA, ET, VG), there is a strong negative correlation between EIS and accuracy, indicating that embeddings with higher interpretability tend to have slightly lower predictive performance. In particular, dimensionality is strongly negatively correlated with accuracy, while sparsity shows a moderate positive correlation and clusterability is weakly or negatively correlated. Conversely, for CV datasets (CIFAR-10, STL-10, Fashion-MNIST), the trends are different: higher EIS and dimensionality often correspond to higher accuracy, highlighting a domain-specific pattern where more interpretable embeddings can also be more predictive. Sparsity and clusterability correlations remain relatively weak and inconsistent across CV datasets. Notably, Fashion-MNIST shows a weak correlation, implying no clear trend between interpretability and accuracy for this dataset. Overall, these results highlight that the relationship between embedding interpretability and downstream performance is domain-dependent.

| Dataset | Pearson (EIS) | Dimensionality | Sparsity | Clusterability |
|---|---|---|---|---|
| CPA | -0.904 | -0.855 | 0.498 | -0.101 |
| ET | -0.808 | -0.380 | 0.091 | -0.684 |
| VG | -0.771 | -0.328 | 0.018 | -0.614 |
| CIFAR-10 | 0.841 | 0.791 | -0.171 | 0.344 |
| STL-10 | 0.797 | 0.741 | -0.097 | 0.353 |
| Fashion-MNIST | 0.093 | 0.331 | -0.530 | 0.158 |

Table 3: Pearson correlations between embedding interpretability measures and downstream classification accuracy for NLP and CV datasets. Negative correlations in NLP datasets indicate that less interpretable embeddings often achieve higher accuracy, while positive correlations in CV datasets suggest that more interpretable embeddings tend to perform better.

## 6 CONCLUSION

In this work, we systematically investigate the trade-offs between interpretability and predictive performance across a wide range of NLP and CV embeddings. By introducing the embedding interpretability score (EIS) and evaluating multiple classifiers on multiple datasets, we provide a unified framework to quantify the relationship between embedding interpretability and accuracy. Our results reveal distinct domain-specific trends: for NLP datasets, higher interpretability generally comes at the cost of predictive performance, consistent with the classical accuracy–interpretability trade-off. In contrast, for CV datasets, more interpretable embeddings often achieve comparable or even higher accuracy, highlighting a favorable alignment between interpretability and performance in vision tasks. Classifier choice has a limited effect in both domains, emphasizing that embedding design is the primary driver of downstream performance.

Beyond technical insights, this framework has broader societal implications. By quantifying embedding interpretability, we enable more transparency, accountability, and trust in machine learning systems, particularly in high-stakes applications such as medical imaging, automated hiring, and financial decision-making. This work encourages the development of models that are not only accurate but also more explainable, supporting the responsible and equitable adoption of machine learning technologies across diverse domains.

**Future Work:** While this study focuses on evaluating the interpretability of embeddings within specific domains, we aim to extend this methodology to quantify and assess the interpretability of the classification models themselves using classification-dependent properties. Cross-domain comparisons introduce additional challenges, as they require incorporating a broader set of metrics capturing model complexity, decision boundaries, and robustness. Furthermore, we plan to expand our framework to support cross-modal analyses, enabling systematic comparisons of interpretability and accuracy between NLP, CV, and potentially other modalities (e.g., speech, multi-modal).

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

# A   APPENDIX

## A.1   EMBEDDING MODELS

Further details about the embedding models employed in this study are provided in Table 4. We include both NLP and CV embeddings, spanning classical word-level models, transformer-based sentence embeddings, and state-of-the-art vision models.

| Acronym | Model Name | Type | Description |
|---|---|---|---|
| W2Vec | Word2Vec | NLP | Classic word embedding model using skip-gram or continuous bag of words to capture semantic and syntactic relationships at the word level. Provides dense, low-dimensional vector representations for vocabulary terms. |
| MiniLM | all-MiniLM-L6-v2 | NLP | Lightweight transformer producing high-quality sentence embeddings efficiently. Captures contextualized semantic meaning with minimal computational overhead. |
| MiniLM-100d | all-MiniLM-L6-v2-100d | NLP | Compact 100-dimensional (using PCA) MiniLM variant; trades off embedding richness for lower computational cost. |
| Para-MiniLM | paraphrase-MiniLM-L6-v2 | NLP | MiniLM fine-tuned on paraphrase detection datasets, producing embeddings that cluster semantically similar sentences tightly. |
| MPNet | all-mpnet-base-v2 | NLP | Transformer combining masked and permuted language modeling for more robust contextual embeddings. Excels at capturing sentence-level semantics. |
| BERT | bert-base-nli-mean-tokens | NLP | Bidirectional transformer pre-trained on large corpora; generates rich contextual sentence embeddings suitable for many downstream NLP tasks. |
| RoBERTa | all-roberta-large-v1 | NLP | Enhanced BERT variant trained with larger datasets and dynamic masking, offering deeper contextual representations and stronger generalization. |
| SetFit-DU | setfit-distiluse-base-multilingual-cased-v2-finetuned-amazon-reviews-multi-binary | NLP | Domain-specific sentence embeddings based on a distilled multilingual transformer. Fine-tuned on Amazon product reviews across multiple languages with a multi-binary classification objective and optimized to cluster semantically similar reviews together. |
| CLIP-ViT-B32 | CLIP Vision Transformer B32 | CV | Multi-modal transformer aligning image and text embeddings; produces image embeddings that capture semantic content aligned with natural language. |
| DINO-ViT-B16 | DINO Vision Transformer B16 | CV | Self-supervised vision transformer learning robust and clusterable image embeddings without labels; captures structural and semantic patterns in visual data. |
| ViT-B16 | Vision Transformer B16 | CV | Transformer-based image model splitting images into 16x16 patches; captures long-range dependencies and global image structure efficiently. |

**Table 4 continued from previous page**

| Acronym | Model Name | Type | Description |
|---|---|---|---|
| DenseNet-121 | DenseNet-121 | CV | Convolutional neural network with dense connectivity; produces expressive feature embeddings but can be less interpretable due to high redundancy. |
| EffNet-B0 | EfficientNet-B0 | CV | Lightweight CNN balancing accuracy, parameter efficiency, and interpretability; scales depth, width, and resolution to optimize embeddings. |
| ResNet-50 | ResNet-50 | CV | Residual network with 50 layers producing high-dimensional embeddings; strong predictive accuracy, but higher dimensionality may reduce interpretability. |

Table 4: Details of the embedding models used in this study.

## A.2 USE OF LARGE LANGUAGE MODELS

Large language models (LLMs) were employed in this study to support multiple aspects of the research. Specifically, they were used to (i) explore, identify, and retrieve relevant prior work, and (ii) assist in polishing and refining the manuscript text. While LLMs guided in language generation and editing, all technical content, analyses, and interpretations presented in this paper were independently developed and verified by the authors.

