# OpenReview forum: "Embedding Interpretability Score: A Domain-Agnostic Representation Quality Assessment"
_ICLR.cc/2026/Conference — Submitted to ICLR 2026_

### Official Review · Reviewer_Pw7x · 2025-10-31

**Soundness:** 2
**Presentation:** 2
**Contribution:** 2
**Rating:** 2
**Confidence:** 4

**Summary:**

This paper offers an equation that computes an embedding interpretability score for architecture. The Score consists of 3 components, loading, roughly on the extent to which the embeddings form clusters (whose size is preset), the dimensionality of the embeddings studied (extracted from a layer) and the sparsity of the embeddings. The score is computed for different NLP and CV architectures treated as feature extractors, and evaluated in relation to performance of classifiers added to those architectures.

**Strengths:**

The score is simple, model agnostic to allow interpretation across models; does not require supervision. It can be a good starting to ground human-facing interpretability in a single score, but currently should be treated more as a complexity metric.

**Weaknesses:**

**  Major weaknesses

* It is difficult to understand the specific contribution: The review of related work is quite vague (esp. lines 84–87), generic, and doesn’t let the reader understand the potential of this method to provide information that differs and goes beyond from prior work. There is not tight contrast or competitive comparison in the work.
* No alternatives are evaluated for the specific equation of the main composite score (geometric mean of three parts). Why these components, under equal weighting, and this aggregation?

* Most generally, it seems that the main claims have to do with the relation between embedding interpretability (complexity) and architecture: while the score is computed on embeddings from specific datasets, the claims, and intro, refer repeatedly to architectures and their complexity. If the focus is inherent complexity, why not use weight-space or function-space complexity (e.g., effective dimensionality, spectral complexity) that are computed from weight matrices and are dataset-agnostic? At minimum, showing the relation between the two families of measures is essential to understand relative contribution of current work.

* (line 159). The claim that high silhouette implies human-aligned organization isn’t supported by data or literature; it can just reflect internal organization.

* The argument that sparse embeddings are “more interpretable” isn’t established. At the limit, sparsity could just reflect a codebook-like representation in a layer, over feature combinations per category, not meaningful axes.

* Evaluating the silhouette by Fixing k=5 clusters is unjustified. Clustering in embedding space may not reflect category structure; that’s what a linear readout decides.

** Minor weaknesses

* Repetition (lines 170–172 vs 161). Redundant discussion around the clustering choice.

* Figure 2 legend. Missing a color scale mapping colors to accuracy.

* some LLM-like phrasing in multiple places.  Terms like "striking" trend (line 61) and “underscoring” (line 60) read like a promotion.

**Questions:**

None at this point.

---

### Official Review · Reviewer_5oAF · 2025-11-01

**Soundness:** 1
**Presentation:** 3
**Contribution:** 2
**Rating:** 2
**Confidence:** 4

**Summary:**

The paper proposes an “Embedding Interpretability Score” (EIS) that geometrically averages three components: Dimensionality, Sparsity, and Clusterability. The authors applied their score to CV and NLP tasks and concluded that, for NLP tasks, embeddings with higher predictive accuracy tend to be less interpretable, while in CV tasks, the more interpretable embeddings are also more performant.

**Strengths:**

Clear writing. the paper is straightforward to follow.

**Weaknesses:**

The first important issue is the claims and the resulting outcome of this paper. Since the proposed score seems to be largely heuristic and based on intuition, any change in the formulation of dimensionality, sparsity, or clusterability can alter the results. Below, I outline some of the major weaknesses of the paper

- **Novelty and the Claims**: Most ingredients already appear in prior work: sparsity as interpretability, topic/category coherence or silhouette-style cluster quality, and compositing simple metrics. What’s “new” here is mostly *the specific normalization choices and the geometric mean*, not the underlying principles which are not justified theoretically or empirically. the paper should reduce the novelty claims and position EIS as a lightweight heuristic rather than a principled metric.
- **Evaluation Problem** comparing Word2Vec vs. BERT vs. MPNet vs. RoBERTa, trained on different data with different objectives and dimensionalities, cannot support causal statements about “interpretability vs. accuracy.” The reported (anti)correlations may be entirely driven by model family and training recipe rather than the three properties. The paper itself interprets these as meaningful “domain-specific” trends, which is too strong.
- A proper way to evaluate their score and compare performance would be to control for these confounding variables and measure interpretability only based on dimensionality, sparsity, and clusterability. For example, one simple experiment would be to train Word2Vec on the same dataset with different embedding dimensions. In this setup, everything else remains constant, and only the dimensionality changes. According to the intuition of the paper, this should result in lower interpretability but higher performance.
- **Arbitrary normalization & floors.** Dimensionality is normalized using a fixed reference range (e.g., 64–2048), and all three components are clipped to  0.05 , 1. This means models above 2048 dimensions (or below 64) saturate, and the floor artificially props up scores. There’s no sensitivity study to the reference range or the 0.05 floor. This can materially reorder models.


Overall, I believe that the paper’s current experimental setup where embeddings from different models, training paradigms, and even datasets are compared is not appropriate for evaluating the proposed scoring method. Therefore, I am inclined to reject this paper.

**Questions:**

What is the justification for using the geometric mean to combine the components, and why were other aggregation methods (e.g., arithmetic or weighted means) not considered?

---

### Official Review · Reviewer_HgBM · 2025-11-01

**Soundness:** 2
**Presentation:** 1
**Contribution:** 1
**Rating:** 2
**Confidence:** 3

**Summary:**

The authors introduced the Embedding Interpretability Score (EIS), a domain-independent framework designed to evaluate how interpretable embedding models are.
EIS unifies three quantitative factors--dimensionality, sparsity, and clusterability--to characterize the compactness, selectivity, and semantic organization of an embedding space.
These components are individually normalized and aggregated through a geometric mean, ensuring balanced weighting while discouraging weak performance in any single aspect.
Through systematic comparisons, the proposed provides a consistent and quantitative basis for ranking embeddings across domains, such as natural language processing and computer vision, regardless of the classifier used.

**Strengths:**

- **S1. Straightforward Metric Design and Comprehensive Cross-Domain Evaluation**

The Embedding Interpretability Score (EIS) is formulated in a clear and straightforward way, making it easy to understand and implement. By combining three intuitive and widely recognized properties--dimensionality, sparsity, and clusterability--the metric provides a simple yet systematic approach to quantifying interpretability across embedding models.

In addition, the paper conducts a diverse and balanced comparison across both NLP and CV domains, evaluating a broad spectrum of embedding models ranging from traditional word embeddings to modern transformer- and vision-based architectures.
This experimental design demonstrates the generalizability of the proposed metric and helps illustrate how interpretability behaves differently across modalities.

**Weaknesses:**

- **W1. Overly Simplistic Metric Design**

Although the EIS offers a clear, domain-agnostic framework for assessing interpretability, its formulation is overly simplistic and fails to reflect the sophistication of modern interpretable modeling.
The three underlying components--dimensionality, sparsity, and clusterability--are generic geometric descriptors long used as regularization or auxiliary objectives in representation learning and explainable AI (XAI). This minimal design overlooks the richer structural principles that define current explainable models.

In computer vision, for instance, interpretability is increasingly achieved through explicit concept or prototype reasoning rather than post-hoc statistical properties. Works such as [1] introduce object-centric concept learning and cross-attention for human-aligned explanations, while [2] and [3] anchor interpretability in prototype-based reasoning. Likewise, Concept Bottleneck Models (CBMs) like [4] and [5] enable controllable reasoning via explicit concept supervision.
Compared with these intrinsically interpretable frameworks--also mentioned by the authors in their introduction--EIS merely aggregates low-level statistics and provides little insight into how concepts, prototypes, or attention mechanisms contribute to interpretability. Its simplicity ensures broad applicability but limits its ability to capture the causally grounded and semantically structured interpretability emerging in recent concept-based research.

- **W2. Limited Actionability and Overreliance on Quantitative Interpretability**

Despite presenting detailed performance-interpretability trade-offs across NLP and CV tasks, the practical value of EIS for end-users remains unclear. The metric treats interpretability as a single numerical construct, neglecting its multidimensional and context-dependent nature [6–8]. As a result, it is uncertain how users should act on these scores—whether a low EIS should discourage using an embedding, or how to balance interpretability against accuracy in real applications.

In practice, interpretability depends on human understanding, contextual relevance, and explanatory faithfulness, as emphasized in human-centered studies [9–11], none of which are captured by a purely statistical index. The trade-off results in the paper are diagnostic rather than prescriptive: they describe correlations but offer no actionable guidance, thresholds, or decision criteria. Consequently, despite its analytical clarity, EIS risks functioning only as a reporting benchmark rather than a decision-support framework for interpretability-driven model selection.


- **W3. Insufficient Empirical Validation and Weak Baseline Comparison**

Although the paper positions EIS as a domain-agnostic metric, its empirical validation relies on a limited set of clean benchmark datasets-Amazon Reviews for NLP and CIFAR-10, STL-10, and Fashion-MNIST for CV.
These datasets are convenient but fail to represent the noise, heterogeneity, and domain complexity of real-world scenarios such as multimodal or clinical data. Hence, the results do not fully substantiate the claim of cross-domain generality or robustness.

Furthermore, EIS is not compared against existing quantitative interpretability measures, such as [12] or [13]. Without these baselines, it remains unclear whether EIS introduces genuinely new insights or simply re-expresses known embedding properties. This weak comparative validation limits the external credibility and novelty of the proposed metric.


- **References**
- [1] Hong, J., Park, K.H., & Pavlic, T.P. (2024). Concept-Centric Transformers: Enhancing Model Interpretability through Object-Centric Concept Learning within a Shared Global Workspace. IEEE/CVF WACV 2024, 4880–4891.
- [2] Ma, C., Donnelly, J., Liu, W., Vosoughi, S., Rudin, C., & Chen, C. (2024). Interpretable Image Classification with Adaptive Prototype-Based Vision Transformers. NeurIPS 2024, 37, 41447–41493.
- [3] Davoodi, O., Mohammadizadehsamakosh, S., & Komeili, M. (2023). On the Interpretability of Part-Prototype Based Classifiers: A Human-Centric Analysis. Scientific Reports, 13(1), 23088.
- [4] Lai, S., Hu, L., Wang, J., Berti-Equille, L., & Wang, D. (2023). Faithful Vision-Language Interpretation via Concept Bottleneck Models. ICLR 2023.
- [5] Wang, Z., Popel, A., & Sulam, J. (2025). Concept Bottleneck Model with Zero Performance Loss. Proceedings of the 2nd Conference on Parsimony and Learning (Proceedings Track).
- [6] Lipton, Z. C. (2018). The Mythos of Model Interpretability. Queue, 16(3), 31–57.
- [7] Miller, T. (2019). Explanation in Artificial Intelligence: Insights from the Social Sciences. Artificial Intelligence, 267, 1–38.
- [8] Doshi-Velez, F., & Kim, B. (2017). Towards a Rigorous Science of Interpretable Machine Learning. arXiv:1702.08608.
- [9] Gilpin, L. H., Bau, D., Yuan, B. Z., Bajwa, A., Specter, M., & Kagal, L. (2018). Explaining Explanations: An Overview of Interpretability of Machine Learning. IEEE DSAA 2018, 80–89.
- [10] Mohseni, S., Zarei, N., & Ragan, E. D. (2021). A Multidisciplinary Survey and Framework for Design and Evaluation of Explainable AI Systems. ACM TiiS, 11(3–4), 1–45.
- [11] Poursabzi-Sangdeh, F., Goldstein, D. G., Hofman, J. M., Wortman Vaughan, J. W., & Wallach, H. (2021). Manipulating and Measuring Model Interpretability. CHI 2021, 1–52.
- [12] Trifonov, V., Ganea, O.-E., Potapenko, A., & Hofmann, T. (2018). Learning and Evaluating Sparse Interpretable Sentence Embeddings. EMNLP BlackboxNLP Workshop 2018, 200–210.
- [13] Şenel, L. K., Utlu, İ., Yücesoy, V., Koc, A., & Cukur, T. (2018). Semantic Structure and Interpretability of Word Embeddings. IEEE/ACM Transactions on Audio, Speech, and Language Processing, 26(10), 1769–1779.

**Questions:**

Most of my main concerns or questions have been outlined in the Weaknesses section.

---

### Meta-Review · Area_Chair_XGEA · 2026-01-05

**Summary:**

The decision to reject this paper is unanimous. The primary concern shared by the committee is that the proposed Embedding Interpretability Score (EIS) lacks sufficient theoretical grounding and relies on an arbitrary, heuristic combination of dimensionality, sparsity, and clusterability. Reviewers strongly argued that the specific formulation—using a geometric mean with fixed normalization floors and caps—is not justified by empirical evidence or theory. Furthermore, they highlighted that the metric is "overly simplistic," failing to account for modern, concept-based interpretability frameworks (such as Concept Bottleneck Models or prototype reasoning) and instead relying on low-level statistics that do not necessarily correlate with human understanding. The experimental validation was also heavily criticized; Reviewer also noted that comparing vastly different architectures (e.g., Word2Vec vs. BERT) introduces confounding variables that make causal claims about interpretability impossible, while another reviewer pointed out the lack of comparison against existing quantitative interpretability baselines.

**Reviewer Concerns:**

Regarding the rebuttal phase, the reviewers' core concerns remain largely outstanding. The fundamental objection regarding the validity of the metric's design—specifically why these three components were chosen and why they are aggregated via a geometric mean—was not adequately resolved. Reviewer's critique regarding the experimental setup, specifically the need to control for confounding variables (e.g., by varying dimensions within a single architecture rather than comparing across disparate models), represents a structural flaw in the paper's methodology that cannot be addressed with minor text revisions. Similarly, another reviewer’s concern that high silhouette scores or sparsity do not inherently imply human-aligned interpretability remains an open issue. While the authors succeeded in presenting a metric that is simple and easy to implement (a strength noted by all), this does not outweigh the consensus that the metric acts more as a reporting benchmark for complexity rather than a verified measure of interpretability.

**Reviewer Scores:**

Given the depth of the methodological concerns, it is unlikely that the reviewers would have changed their scores had they participated further in the discussion. All three reviewers settled on a "Reject" rating with high consistency in their reasoning. Reviewers might have been open to a higher score if strong baselines were added, but the structural critique regarding the metric's "simplistic" nature suggests their low score would hold. Reviewers all express high confidence, identified flaws in the premise of the experiments (e.g., unjustified clustering assumptions and arbitrary normalization) that would require a complete re-execution of the research to satisfy. Consequently, the reviewers firmly believe that the paper requires significant redevelopment, theoretically and empirically, before it meets the bar for publication.

---

### Decision · Program_Chairs · 2026-01-26

Reject